# *ZmNAC17* Regulates Mesocotyl Elongation by Mediating Auxin and ROS Biosynthetic Pathways in Maize

**DOI:** 10.3390/ijms25094585

**Published:** 2024-04-23

**Authors:** Ran Yang, Kangshi Li, Ming Wang, Meng Sun, Qiuhua Li, Liping Chen, Feng Xiao, Zhenlong Zhang, Haiyan Zhang, Fuchao Jiao, Jingtang Chen

**Affiliations:** 1College of Agronomy, Qingdao Agricultural University, Qingdao 266109, China17860727572@163.com (K.L.); 15224753053@139.com (Q.L.); lipingchen1203@163.com (L.C.); m17860729158@163.com (F.X.);; 2The Characteristic Laboratory of Crop Germplasm Innovation and Application, Provincial Department of Education, College of Agronomy, Qingdao Agricultural University, Qingdao 266109, China

**Keywords:** *ZmNAC17*, maize, mesocotyl, auxin, ROS

## Abstract

The mesocotyl is of great significance in seedling emergence and in responding to biotic and abiotic stress in maize. The NAM, ATAF, and CUC2 (NAC) transcription factor family plays an important role in maize growth and development; however, its function in the elongation of the maize mesocotyl is still unclear. In this study, we found that the mesocotyl length in *zmnac17* loss-of-function mutants was lower than that in the B73 wild type. By using transcriptomic sequencing technology, we identified 444 differentially expressed genes (DEGs) between *zmnac17-1* and B73, which were mainly enriched in the “tryptophan metabolism” and “antioxidant activity” pathways. Compared with the control, the *zmnac17-1* mutants exhibited a decrease in the content of indole acetic acid (IAA) and an increase in the content of reactive oxygen species (ROS). Our results provide preliminary evidence that *ZmNAC17* regulates the elongation of the maize mesocotyl.

## 1. Introduction

The mesocotyl is of great significance in seedling emergence, root development, and stress resistance. Maize seedlings with long mesocotyls have stronger germination ability and higher rates of seedling emergence [1]. The mesocotyl cortical aerenchyma supports shoot-borne crown roots [2,3], and its parenchyma plays a substantial role in regulating stress signal transduction in the xylem [4]. Various genes have been reported to play critical roles in maize mesocotyl elongation. Chen et al. (2023) reported that *ZmCOP1* mediates mesocotyl elongation through the brassinosteroid signaling pathway [5]. Moreover, it has been found that *ZmMYB59* negatively regulates mesocotyl elongation through the gibberellin signaling pathway [6]. Zhang et al. (2023) found that the overexpression of *ZmWRKY28* in maize leads to an increase in mesocotyl length under dark conditions [7].

Auxin levels and distribution influence growth and development in plants [8], and previous studies have suggested that auxin also directly or indirectly promotes the elongation of maize mesocotyl. For example, *ZmLA1* (a functional ortholog of *LAZY1* in rice) affects maize mesocotyl elongation in the dark [9], and coronatine inhibits auxin synthesis by increasing the release of endogenous ethylene, thereby suppressing cell elongation and subsequently reducing the length of the mesocotyl [10]. In line with this, auxin-binding protein 1 (ABP1) has also been found to promote mesocotyl elongation in maize [11]. 

Reactive oxygen species (ROS) are involved in plant growth, hormone signaling, and biotic and abiotic stress responses [12,13]. ROS production and processing are linked to NADPH oxidase, superoxide dismutase (SOD), class III peroxidase (POD), etc. [14], where POD participates in the formation of maize lignin, which is related to cell wall hardening and mesocotyl elongation [15]. Additionally, ROS affect auxin metabolism, distribution, transport, and signal transduction [16]. Interestingly, based on previous reports, auxin also participates in ROS production, and the dynamic equilibrium between auxin and ROS is related to the redox balance [17,18,19]. 

The NAM, ATAF1,2, and CUC2 (NAC) transcription factor family influences various growth and developmental processes in plants, including seed development, hormone signal transduction, crop yield and quality, etc. [20,21,22,23]. Xie et al. (2000) reported that *AtNAC1* promotes lateral root development by transducing the auxin signal downstream of transport inhibitor response 1 (TIR1) [24]. In rice, *OsNAC2* affects plant height and flowering time by mediating the gibberellic acid pathway [25]. *AtRD26* (an NAC transcription factor) enhances stress resistance in post-harvest fruits by increasing the expression of stress-related genes [22]. *Arabidopsis thaliana* NAC transcription factor 1 (ATAF1) binds to and regulates the *9-cis-epoxycarotenoid dioxygenase 3* (*NCED3*) gene, which is responsible for ABA biosynthesis [26]. NAC transcription factors can negatively regulate leaf senescence by influencing the contents of salicylic acid (SA) and ROS [27]. The NAC transcription factor, NAC with transmembrane motif 1-like (NTL4), during drought stress, promotes the production of ROS by directly binding to the promoters of genes related to ROS biosynthesis and accelerates leaf senescence [28]. Another NAC transcription factor, SNAC, has been reported to enhance tolerance to high temperatures, drought, and oxidative stress in rice by regulating the dynamic balance of ROS [29].

Despite the fact that extensive research has been conducted on the NAC family, its relationship with mesocotyl elongation has not been thoroughly studied. In this study, we analyzed the phenotypic characteristic of two *zmnac17* loss-of-function mutants in maize and identified DEGs between *zmnac17-1* and B73. We proposed that *ZmNAC17* positively regulates the elongation of the maize mesocotyl by mediating the auxin and ROS synthetic pathways.

## 2. Results

### 2.1. Phenotype of Maize zmnac17-1 and ZmNAC17-2 Mutants

In order to explore the effect of NAC transcription factors on the length of the maize mesocotyl, we collected a loss-of-function *zmnac17-1* mutant from the EMS-induced mutant library, denoted as *zmnac17-1*. This was a start-loss mutant featuring a change in nucleotides from ATG to ATA (Appendix A).

We measured the mesocotyl length of *zmnac17-1* specimens, using B73 as a control (Figure 1A). After having been grown in the dark for 7 days, B73 presented a mesocotyl length of about 9.30 cm, and *zmnac17-1* of about 8.00 cm; thus, the latter showed significantly shorter seedling length than B73 (Figure 1B). In addition, the mesocotyl fresh weight was about 0.30 g in B73, while in *zmnac17-1*, it was about 0.23 g, a significantly lower value (Figure 1C). In order to ensure the accuracy of this experiment, we collected another loss-of-function *zmnac17* mutant, denoted as *zmnac17-2*, a stop-gain mutant featuring a change in nucleotides from TGG to TGA at the 543rd nucleotide (Appendix A). We measured the length and fresh weight of the mesocotyl in *zmnac17-2* and obtained 8.40 cm and 0.25 g, respectively. Both values were significantly lower than their counterparts in B73 (Appendix A). 

### 2.2. GO and KEGG Enrichment Analyses

In order to gain further insights into which genes were differentially expressed, we conducted RNA-seq analyses using *zmnac17-1* because of both its start-codon lost genotype and its more pronounced phenotype (Appendix A). In total, 148 genes were up-regulated and 296 genes down-regulated in *zmnac17-1* compared with B73 (Figure 2A,B). We also performed GO and KEGG enrichment analyses to determine which processes were affected. The GO enrichment analysis showed that the DEGs were enriched in oxidoreductase activity and aldehyde oxidase activity in terms of molecular functions, and in the abscisic acid metabolic process, auxin biosynthetic process, and tryptophan metabolic process in terms of biological processes (Figure 2C, Appendix A). 

The KEGG enrichment analysis showed that the DEGs were enriched in plant hormone signal transduction, tryptophan metabolism, plant–pathogen interaction (which has a tight relationship with ROS homeostasis), etc. (Figure 2D, Appendix A). In conclusion, the GO and KEGG analyses showed that the DEGs were related to plant hormone signal transduction and the biosynthesis of metabolites. Therefore, we speculate that these pathways may play an important role in how *ZmNAC17* regulates mesocotyl length.

### 2.3. Auxin Regulates Mesocotyl Length in zmnac17-1 Mutant

Auxin is a critical regulator of mesocotyl length [30]. In plants, the biosynthesis of auxin includes two pathways: a tryptophan-dependent one and a tryptophan-independent one [31]. In this study, we identified various tryptophan metabolism-related DEGs (Table 1). The qRT-PCR results also confirmed the reliability of RNA-seq data (Figure 3A). Four genes (Zm00001eb396770, Zm00001eb396780, Zm00001eb396760 and Zm00001eb396790) were expressed at a lower level in the *zmnac17-1* mutant, while Zm00001eb416690 was expressed at a higher level in the *zmnac17-1* mutant (Table 1, Figure 3A). 

Thus, we propose that the transcriptional changes in the auxin-related genes may partially explain the short mesocotyl phenotype of the *zmnac17-1* mutant. In order to verify whether *ZmNAC17* regulates the biosynthesis of auxin, we measured the IAA content in *zmnac17-1* and B73 (Figure 3B) and found that B73 presented about 13.97 ng/g IAA and *zmnac17-1* about 10.70 ng/g IAA; that is, the content of IAA was significantly lower (by 23.5%) in *zmnac17-1*. This result verifies that auxin participates in regulating mesocotyl elongation. The KEGG enrichment analysis showed that the DEGs between *zmnac17-1* mutant and B73, such as *ZmSAUR54* (Zm00001eb241870), *ZmSAUR11* (Zm00001eb052590), and *ZmIAA32* (Zm00001eb301590) (Figure 3C), were enriched in auxin signal transduction pathways. Taken together, our results show that the DEGs relative to tryptophan metabolism and auxin signal transduction may be the reason for the decrease in IAA content.

### 2.4. ROS Level Change in zmnac17-1 Mutant during Mesocotyl Elongation

Based on a previous study, a change in auxin homeostasis could lead to altered ROS levels in plants [19]. Moreover, it has been reported that the length of the maize mesocotyl is negatively correlated with H_2_O_2_ content and POD activity [15]. Interestingly, in this study, we also found that the DEGs were enriched in terms of oxidoreductase activity, hydrogen peroxide catabolic process, and antioxidant activity. This result implies that the ROS level may have changed during mesocotyl elongation in our specimens. qRT-PCR was used to confirm the expression patterns of the DEGs related to antioxidant activity (Table 2, Figure 4). Based on our results, we speculate that the shortened mesocotyl length in *zmnac17-1* may be related to the change in ROS content. We measured the total ROS content and found that it was about 252.40 ng/mL in B73 and about 147.70 ng/mL in *zmnac17-1*; i.e., the content of ROS was significantly higher (by 41%) in the mutant (Figure 5A). 

The content of ROS in plants is closely related to glutathione (GSH), oxidized glutathione (GSSG), catalase (CAT), POD, and SOD, among other factors [14]; therefore, we tested the activity of these enzymes. The results showed that GSH was present at 0.65 umol/g in B73 and 0.75 umol/g in *zmnac17-1*; i.e., the content of GSH was 15% higher in *zmnac17-1*. The GSSG content was 82.77 nmol/g in B73 and only about 40.61 nmol/g in *zmnac17-1*, representing a 51% reduction in this parameter in the mutant. In addition, the ratio of GSH/GSSG was higher in *zmnac17-1* than in B73. We also measured the contents of CAT, POD, SOD, and MDA. Interestingly, we found that CAT activity was 185.5 umol/min/g in B73, but only about 150.00 umol/min/g in *zmnac17-1*, i.e., we observed a 19% reduction in this parameter in the mutant. POD activity was significantly lower in B73 than in the *zmnac17-1* mutant. SOD activity was about 247.50 U/g in B73, and about 234.60 U/g in *zmnac17-1*. MDA content was about 4.30 nmol/g in B73, and about 4.53 nmol/g in *zmnac17-1*, with no significant difference between B73 and *zmnac17-1* (Figure 5B–H). In summary, these results indicate that changes in ROS levels may be responsible for the short mesocotyl phenotype. 

## 3. Discussion

Based on the study conducted by Fan et al. in 2014 on *Arabidopsis thaliana*, *ANAC074* is a homologue gene of *ZmNAC17* [21]. Interestingly, *ANAC074* has been found to be related to the secondary tissue differentiation of the hypocotyl [32]. However, the specific function of *ANAC074* in the latter has not been reported. In this study, we showed that *ZmNAC17* is involved in mesocotyl elongation in etiolated maize seedlings (Figure 1A and Appendix A). The etiolated seedlings of the *zmnac17-1* mutant exhibited a shorter mesocotyl compared with B73 when grown in the dark.

In this study, the content of IAA was significantly decreased in *zmnac17-1*, indicating that the auxin metabolism pathway was different in the mutant. The RNA-seq analysis identified some DEGs between the *zmnac17-1* mutant and B73, such as *ZmSAUR54* (Zm00001eb241870), *ZmSAUR11* (Zm00001eb052590), and *ZmIAA32* (Zm00001eb301590). We confirmed the lower expression of Zm00001eb396770, Zm00001eb396780, Zm00001eb396760, and Zm00001eb396790 in *zmnac17-1* using qRT-PCR. Since these genes are involved in tryptophan metabolism, the lower expression of these genes may be the reason for the decrease in IAA content. Previous research has shown that auxin promotes mesocotyl elongation [33]. In *Arabidopsis thaliana*, several genes encoding AUX/IAA and SAUR proteins are associated with hypocotyl elongation [34,35]. Moreover, studies have reported that NAC transcription factors regulate plant growth and development by inducing auxin synthesis [36,37]. For example, in *Arabidopsis*, *AtNAC1* promotes lateral root growth by activating the expression of two auxin-related genes, DNA-binding protein (DBP) and auxin-induced in root cultures 3 (AIR3) [24]. *OsNAC2* regulates root development by regulating the auxin and cytokinin signaling pathways [38]. Therefore, we infer that *ZmNAC17* could regulate auxin-related genes, induce the auxin synthesis and signaling pathway, and affect mesocotyl elongation. 

Previous studies have shown that *NAC075* directly up-regulates the expression of catalase 2 (CAT2) and inhibits the accumulation of ROS in *Arabidopsis thaliana* [39]. Similarly, we found that CAT activity was lower, while ROS content was higher, in *zmnac17-1* than in B73. Zhao et al. (2022) found that when H_2_O_2_ accumulates in the maize mesocotyl after light stimulation, POD-induced lignin monomer oxidizes to form lignin, which leads to cell wall hardening and inhibits mesocotyl elongation [15]. The POD activity in the *zmnac17-1* mutant was significantly higher than in B73, which was consistent with Zhao’s study. In addition, the RNA-seq results showed that among the antioxidant activity-related genes, four (Zm00001eb330530, Zm00001eb333290, Zm00001eb111420, and Zm00001eb348950) were down-regulated, and two (Zm00001eb226370 and Zm00001eb109910) were up-regulated. Therefore, we speculate that changes in the expression of these genes could affect ROS levels and be regulated by *ZmNAC17*.

Changes in plant growth and development are often associated with the dynamic equilibrium between auxin and ROS [19]. Mangano et al. (2017) found that there is a molecular link between auxin and ROS-mediated polar root hair growth [17]. Previous research has reported that ROS induce auxin [40]. However, in this study, the auxin content decreased while the ROS content increased in the *zmnac17-1* mutant. More research needs to be carried out to reveal the function of *ZmNAC17* in the auxin–ROS imbalance. In addition, considering that the growth response of plants under environmental stress is affected by the interaction between these two elements [19], we speculate that *ZmNAC17* may also affect growth and development in maize under environmental stress.

Based on the above results and previous studies, a possible molecular network of *ZmNAC17*’s involvement in mesocotyl elongation in maize was constructed (Figure 6). Briefly, the findings of this study indicate that *ZmNAC17* may play a positive regulatory role in mesocotyl elongation under dark growing conditions and that this may be involved in the metabolism of endogenous auxin and ROS. Since the mutants we used were EMS-induced, there might be some unknown mutation sites affecting the elongation of the mesocotyl. It is important to confirm the phenotype by using CRISPR gene knock-out lines. The functional verification of the *ZmNAC17* gene in maize and the identification of genes downstream of *ZmNAC17* are also important directions for future research. Our findings thus provide useful resources for gene discovery and functional identification of seed germination in production.

## 4. Materials and Methods

### 4.1. Plant Materials and Phenotypic Analysis

The *zmnac17-1* and *zmnac17-2* mutants were collected from the maize EMS mutant library [41]. The B73 wild type from the same library was used as a control. All materials were propagated in Jiaozhou, Shandong (36°27′ N, 120°03′ E) in summer and in Ledong, Hainan (18°45′ N, 109°10′ E) in winter. Genotyping was evaluated using Sanger sequencing. DNA from leaves was extracted using the CTAB method. Primers were designed using the NCBI website (Appendix A). PCR was performed using 2× Taq PCR StarMix with Loading Dye (A012, GenStar). Sanger sequencing was performed by Sangon Biotech Co., Ltd (Shanghai, China).

*zmnac17-1*, *zmnac17-2*, and B73 were used for mesocotyl elongation analysis. *zmnac17-1* and B73 were used for physiological and biochemical analyses, hormone content determination, and RNA sequencing analysis. Seeds were sown in a 54 × 28 × 9 cm high-footed seedling tray in the dark. Each hole was filled with vermiculite and then fully watered, and the seedlings were grown in a dark incubator (25 °C). The mesocotyl lengths of the 7-day-old seedlings were measured according to previously published methods [5]. At least 15 individual seedlings for each genotype were analyzed.

### 4.2. Determination of ROS and Antioxidant Metabolites

In this study, seven-day-old *zmnac17-1* and B73 seedlings were used. The experiment was carried out according to the instructions of the plant reactive oxygen species (ROS) ELISA kit (mlROS-96), which was purchased from Shanghai Enzyme-Linked Biotechnology Co., LTD. (Shanghai, China). GSH, GSSG, POD, SOD, CAT, and MDA contents were detected using assay kits (Art. No. G0206W; Art. No. G0207W; Art. No. G0107W; Art. No. G0101W; Art. No. G0105W; and Art. No. G0109W), which were purchased from Suzhou Grace Bio-technology Co. LTD., Suzhou, China. 

### 4.3. Determination of Endogenous IAA

Seven-day-old *zmnac17-1* and B73 seedlings were used for endogenous IAA determination. The experiment was performed by Wuhan MetWare Biotechnology Co., Ltd. (Wuhan, China). 

Fresh seedlings were ground into powder with a MM400 mortar grinder (Retsch, 30 Hz, 1 min). Then, 50 mg powder was dissolved in 1 mL methanol/water/formic acid (15:4:1, *v*/*v*/*v*) and 10 μL internal standard solution (Olchemim/isoReag). The mixture was vortexed for 10 min and centrifuged for 5 min (12,000 rpm, 4 °C). The supernatant was transferred to a new centrifuge tube, evaporated, and dissolved in 100 μL of methanol (80%, *v*/*v*), then filtered with a 0.22 μm membrane.

An UPLC-ESI-MS/MS system (ExionLC™ AD; QTRAP^®^ 6500+) was used for sample extracts analysis. The analytical conditions were as follows: column, Waters ACQUITY UPLC HSS T3 C18 (100 mm × 2.1 mm i.d., 1.8 µm); solvent system, (A) water with 0.04% acetic acid, (B) acetonitrile with 0.04% acetic acid; gradient program, 5% B (0–1 min), 95% B (1–8 min), 95% B (8–9 min), 5% B (9.1–12 min); flow rate, 0.35 mL/min; temperature, 40 °C; injection volume, 2 μL. The ESI source operation parameters were as follows: ion source, ESI+/−; source temperature, 550 °C; ion spray voltage, 5500 V (Positive), −4500 V (Negative); curtain gas, 35 psi. 

Multiple reaction monitoring (MRM) was used for IAA content analysis. The parameters for IAA were Q1 176.1 Da; Q3 130.1 Da; Rt 5.12 min; declustering potential 20; and collision energy 20. Analyst 1.6.3 software (Sciex, Framingham, MA, USA) was used for data acquisition. Multiquant 3.0.3 software (Sciex) was used for IAA quantification. 

### 4.4. RNA Sequencing and Data Analysis

The seven-day-old *zmnac17-1* and B73 seedlings in four independent biological replicates were further used for RNA sequencing analysis. Total RNA extraction, mRNA library construction, and data analysis were performed by Wuhan MetWare Biotechnology Co., Ltd. (Wuhan, China). 

The qualified mRNA library was sequenced using the Illumina NovaSeq 6000 sequencing platform. The sequencing reads were mapped to the maize reference genome (Zm-B73-REFERENCE-NAM-5.0) using HISAT software http://ccb.jhu.edu/software.shtml accessed on 15 April 2024 [42]. StringTie and featureCounts were used for gene annotation and FPKM calculation [43]. DESeq2 was used to analyze the expression-related differences between B73 and *zmnac17-1*. Genes with false discovery rate (FDR) < 0.05 and |log2 Fold Change| ≥ 1 were considered differentially expressed genes (DEGs).

### 4.5. qRT-PCR Analysis

A SteadyPure plant RNA extraction kit (AG21019; Accurate Biotechnology, Co., Ltd., Changsha, China) was used to extract total RNA. An Evo M-MLV RT Mix Kit (AG11728) was used for reverse transcription. The SYBR^®^ Green Premix Pro Taq HS qPCR Tracking Kit (AG11733) and an ABI 7500 Real-Time PCR System were used for fluorescence quantification. The relative gene expression was calculated with the 2^−∆∆Ct^ method, and *Actin* was used as an internal control [44]. The primers used for qRT-PCR can be found in Appendix A.

### 4.6. Statistical Analysis

Student’s unpaired *t*-test was performed using SPSS (version 23.0) and GraphPad Prism8 software (version 8.0.2; GraphPad Software, San Diego, CA, USA).

## Figures and Tables

**Figure 1 ijms-25-04585-f001:**
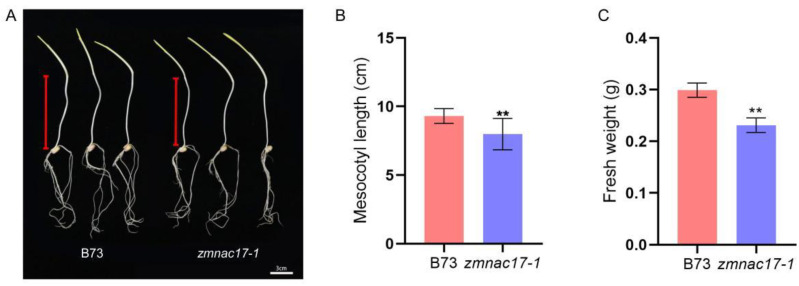
Phenotypic analyses of B73 and *zmnac17-1* (mutant). (**A**) Seedling phenotypes of B73 and *zmnac17-1* grown in the dark for 7 days. Red lines: mesocotyl. Bars: 3 cm. (**B**) Results of quantification of mesocotyl length. (**C**) Results of quantification of mesocotyl fresh weight. Data are means ± SDs of at least 10 biological replicates. Statistical analysis conducted using Student’s unpaired *t*-test (** *p* < 0.01).

**Figure 2 ijms-25-04585-f002:**
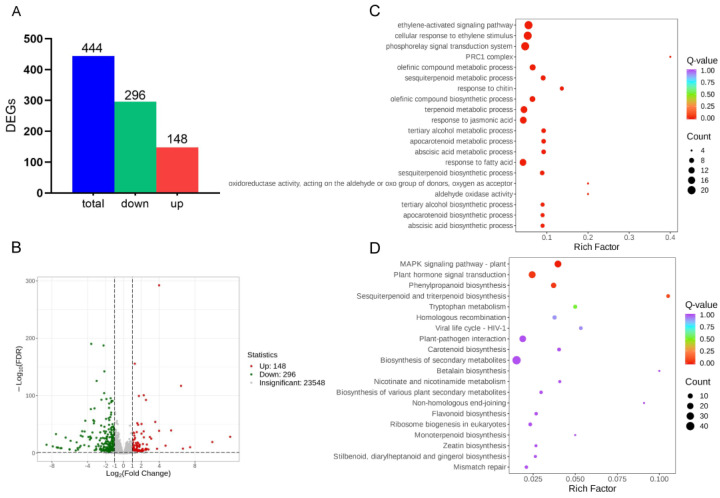
The RNA-seq results show that the *zmnac17-1* mutation led to expression changes in some genes. (**A**) The differentially expressed genes between the B73 wild type and the *zmnac17-1* mutant. (**B**) DEG volcano map, where the red dots represent the up-regulated genes in *zmnac17-1*, the green dots represent the down-regulated genes, and the gray dots represent the non-differentially expressed genes. (**C**) GO enrichment top 20 scatter plot, where the ordinate represents the GO entry, and the abscissa represents the rich factor. (**D**) KEGG enrichment top 20 scatter plot, where the ordinate represents the KEGG path, and the abscissa represents the rich factor.

**Figure 3 ijms-25-04585-f003:**
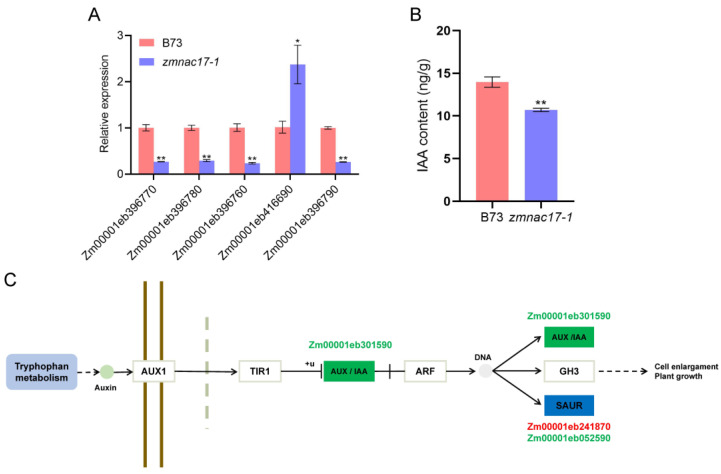
DEGs related to tryptophan metabolism and IAA content in B73 and *zmnac17-1* mutant. (**A**) Verification of DEGs with RT-qPCR. (**B**) IAA content in B73 and *zmnac17-1*. Three biological replicates were used. (**C**) Auxin signaling pathway. Substances in blue boxes represent up-regulated genes, while those in green boxes represent down-regulated genes. Data presented are means ± SDs. Statistical analysis conducted by using Student’s unpaired *t*-test (** *p* < 0.01).

**Figure 4 ijms-25-04585-f004:**
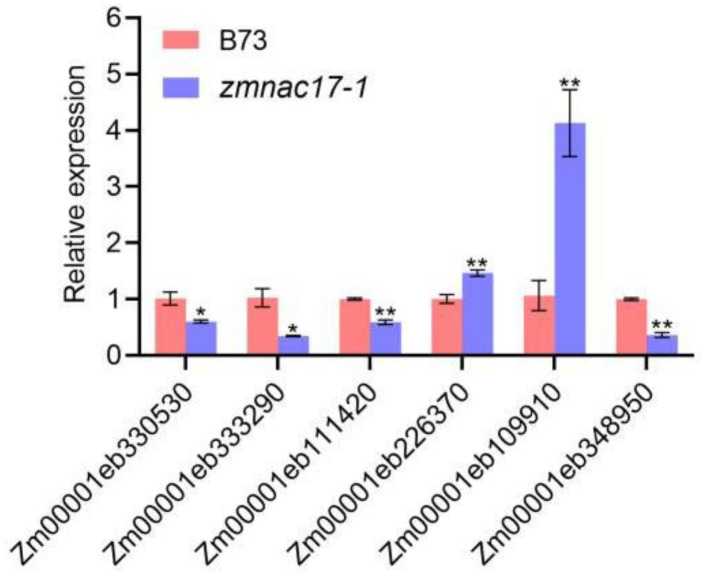
Verification of DEGs related to antioxidant activity with RT-qPCR (* *p* < 0.05 and ** *p* < 0.01).

**Figure 5 ijms-25-04585-f005:**
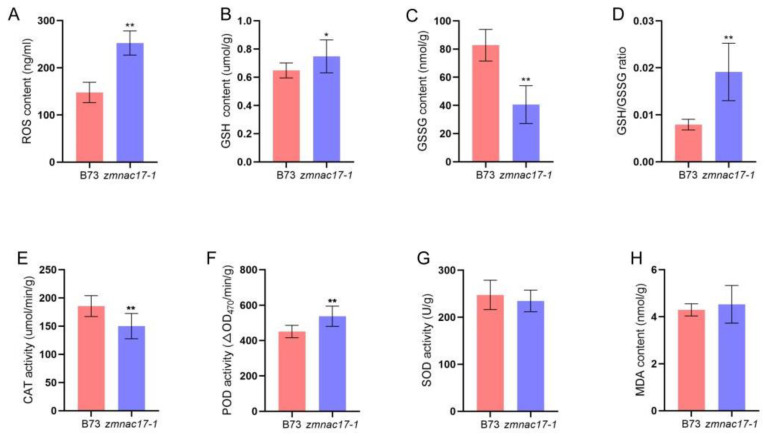
ROS content in B73 and *zmnac17-1* mutant. (**A**–**H**) ROS, GSH, GSSG, GSH/GSSG ratio, CAT, POD, SOD, and MDA content in B73 and *zmnac17-1*. Data are means ± SDs of at least 5 biological replicates. Statistical analysis conducted using Student’s unpaired *t*-test (* *p* < 0.05 and ** *p* < 0.01).

**Figure 6 ijms-25-04585-f006:**
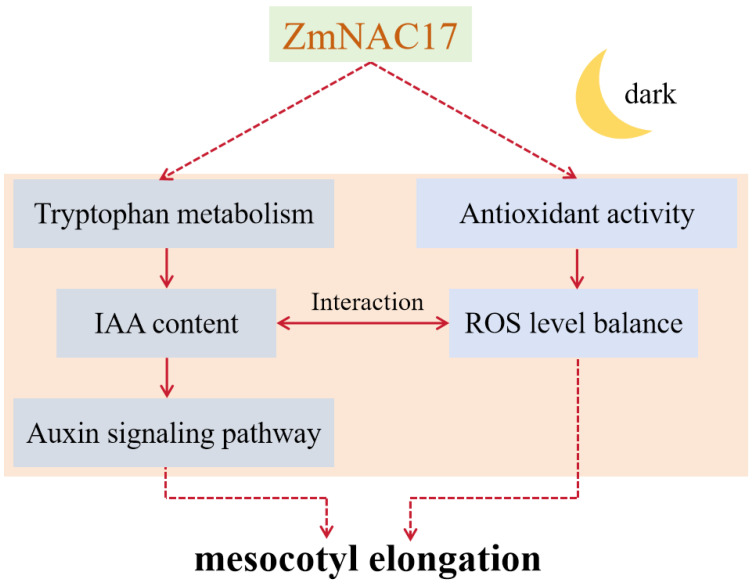
Molecular network of ZmNAC17’s involvement in mesocotyl elongation. The solid line represents what has been experimentally verified [39], and the dotted line represents what is hypothetical.

**Table 1 ijms-25-04585-t001:** DEGs related to tryptophan metabolism.

Gene ID	Annotation	B73_fpkm	*zmnac17-1*_fpkm	Log_2_FC
Zm00001eb396770	Aldehyde oxidase AA01 (possible homologue gene in Arabidopsis)	11.89	0.80	−3.920
Zm00001eb396780	E3 ubiquitin-protein ligase UPL6	7.95	0.60	−3.76
Zm00001eb396760	Alpha hydrolases-like domain-containing protein (possible homologue gene in Arabidopsis)	3.43	0	−7.34
Zm00001eb416690	Tryptophan decarboxylase 1-like	0.11	0.64	2.45
Zm00001eb396790	Indole-3-acetaldehyde oxidase	2.85	0.31	−3.22

**Table 2 ijms-25-04585-t002:** DEGs related to antioxidant activity.

Gene ID	Annotation	B73_fpkm	*zmnac17-1*_fpkm	Log_2_FC
Zm00001eb330530	peroxidase 70 isoform X1	25.90	12.72	−1.06
Zm00001eb333290	peroxidase 72 precursor	18.69	9.29	−1.04
Zm00001eb111420	peroxidase 66 precursor	5.79	2.61	−1.18
Zm00001eb226370	Peroxidase 45	2.22	4.78	1.07
Zm00001eb109910	peroxidase 2-like	0.08	1.09	3.76
Zm00001eb348950	peroxidase 2	0.68	0.15	−2.20

## Data Availability

The raw sequence data reported in this study were deposited in the Genome Sequence Archive (Genomics, Proteomics and Bioinformatics 2021), National Genomics Data Center (Nucleic Acids Res 2022), China National Center for Bioinformation/Beijing Institute of Genomics, Chinese Academy of Sciences (GSA: CRA015622), and are publicly accessible at https://ngdc.cncb.ac.cn/gsa.

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
