# Peer review of "ZmNAC17 Regulates Mesocotyl Elongation by Mediating Auxin and ROS Biosynthetic Pathways in Maize"

_ijms, 2024, doi:10.3390/ijms25094585_

Round 1

Reviewer 1 Report

Comments and Suggestions for Authors

The authors' study on mesocotyl elongation through mediating the auxin and ROS biosynthetic pathways in maize is significant for plant breeders. However, there are some major suggestions for the authors:

 1.       I suggest authors to utilize known genes as references for gene expression analysis with DEGs identified in their study. This will provide a more comprehensive understanding of gene expression patterns and function of Auxin inside the plant for mesocotyl elongation in maize.

 2.       The authors suggested to create a pathway diagram to visually represent the interactions between NAC17, the auxin and ROS biosynthetic pathways. This diagram would enhance the readability and clarity of the study for readers.

 3.       This is very important point how endogenous auxin expression was measured in both mutant and control plants?. Additionally, the authors suggested to use use exogenous auxin to assess the function of the ZmNAC17 gene and its expression in both mutants and control plants. This would provide valuable insights into the role of auxin in mesocotyl elongation.

Author Response

Thank you very much for taking the time to review this manuscript. Please find the detailed responses below and the corresponding revisions highlighted in Red in the re-submitted files.

The authors' study on mesocotyl elongation through mediating the auxin and ROS biosynthetic pathways in maize is significant for plant breeders. However, there are some major suggestions for the authors:

1. I suggest authors to utilize known genes as references for gene expression analysis with DEGs identified in their study. This will provide a more comprehensive understanding of gene expression patterns and function of Auxin inside the plant for mesocotyl elongation in maize.

Thanks for the suggestion. We fully agree with the suggestion. Therefore, we have searched for known genes involved in tryptophan metabolism and antioxidant activity and listed them in Tables 1 and 2, respectively. Please find lines 144 (Table 1) and 165 (Table 2) in the revised manuscript.

2. The authors suggested to create a pathway diagram to visually represent the interactions between NAC17, the auxin and ROS biosynthetic pathways. This diagram would enhance the readability and clarity of the study for readers.

Thanks for your suggestion. We fully agree with the suggestion. Therefore, we created a pathway diagram at the end of the manuscript. Please find lines 247-250 (Figure 6) in the revised manuscript.

3. This is very important point how endogenous auxin expression was measured in both mutant and control plants?. Additionally, the authors suggested to use use exogenous auxin to assess the function of the ZmNAC17 gene and its expression in both mutants and control plants. This would provide valuable insights into the role of auxin in mesocotyl elongation.

Thanks for the critical comments. We fully agree with the suggestion. Therefore, we have described the measurement method for endogenous auxin expression in lines 281-292 in the revised manuscript.

We appreciate your suggestion. Both auxin and ROS were affected in the zmnac17-1 mutant. We proposed that ZmNAC17 plays a role in the imbalance of auxin and ROS, as shown in lines 248-250 (Figure 6). The relationship between auxin content and  the elongation of maize mesocotyl might be complicated. Currently, we are generating ZmNAC17 overexpression and CRISPR knockout transgenic lines. In the future, we would like to use transgenic lines to study the function of ZmNAC17 using both exogenous auxin and ROS. Can we leave this question open in this manuscript? 

Reviewer 2 Report

Comments and Suggestions for Authors

The manuscript by Yang et al., describes the role of ZmNAC17 transcription factor in mesocotyl elongation in Zea mays, using phenotypic and transcriptomic comparison between wild-type and mutant lines. The study is complemented by auxin and ROS content measurements and analysis of anti-oxidant related enzyme activity. As the authors state, their results provided preliminary evidence that ZmNAC17 regulates the elongation of maize mesocotyl. In order to improve the quality of presentation, I suggest the following:

- the authors should provide a schematic represention of the gene (ZmNAC17), depicting the 2 mutations  (17-1 and 17-2) (e.g. in Fig.1 or in a Suppl. Figure). They should also explain the difference of these two mutants and why they had chosen the 17-1 for RNAseq and biochemical analysis

- the statistic data concerning the RNAseq, such as total reads, % mapped genes, total number of genes etc. should be provided in a Table or Suppl. Figure. Further, the authors have to state the public database where the raw transcriptomic data of their RNAseq experiment have been deposited.

- in Line 130 and Line 145 is stated that KEGG enrichment analysis showed that the DEGs are enriched in auxin signal transduction pathway, tryptophan metabolism, oxidoreductase activity, hydrogen peroxide catabolic process and antioxidant activity. However, in Figure 2C and D (i.e., the top 20 in GO and KEGG analysis) only two of the above categories are shown. The authors should provide as a Suppl. Table all the enriched categories of their GO analysis

-  in Lines 150-152 the authors say that total ROS content was lower in the mutant compared to wild-type, in contrast to Figure 5A (the opposite is also stated in Lines 202-203). Additionally, I believe that the data in Lines 157-170 could be provided as a Table or Figure; giving all theses information in text may be tiring and confusing for the readers

- the authors should rewrite Lines 220-228 in order to be more comprehensive and to avoid misunderstanding. I also suggest that Suppl. Figure 2 could be a main figure and (as a model) to be more discussed. 

Comments on the Quality of English Language

Minor editing of English language required

Author Response

Thank you very much for taking the time to review this manuscript. Please find the detailed responses below and the corresponding revisions highlighted in Red in the re-submitted files.

The manuscript by Yang et al., describes the role of ZmNAC17 transcription factor in mesocotyl elongation in Zea mays, using phenotypic and transcriptomic comparison between wild-type and mutant lines. The study is complemented by auxin and ROS content measurements and analysis of anti-oxidant related enzyme activity. As the authors state, their results provided preliminary evidence that ZmNAC17 regulates the elongation of maize mesocotyl. In order to improve the quality of presentation, I suggest the following:

- the authors should provide a schematic represention of the gene (ZmNAC17), depicting the 2 mutations  (17-1 and 17-2) (e.g. in Fig.1 or in a Suppl. Figure). They should also explain the difference of these two mutants and why they had chosen the 17-1 for RNAseq and biochemical analysis

Thanks for the comments. We fully agree with the suggestion. As suggested, we have added a schematic representation of ZmNAC17, depicting the 2 mutations (zmnac17-1 and zmnac17-2), please find lines 460-463 (Supplementary Figure 1).

As suggested, we have described the difference of the mutations in lines 78-79 and lines 86-88 in the revised manuscript.

We chosen the zmnac17-1 mutant because of both its start-codon lost genotype and its more pronounced phenotype, as described in lines 99-100.

- the statistic data concerning the RNAseq, such as total reads, % mapped genes, total number of genes etc. should be provided in a Table or Suppl. Figure. Further, the authors have to state the public database where the raw transcriptomic data of their RNAseq experiment have been deposited.

Thank you for your suggestion. We fully agree with the suggestion. As suggested, we have added the data in Supplementary table 2.

The raw sequence data reported in this study were deposited in the Genome Sequence Archive (Genomics, Proteomics & Bioinformatics 2021), National Genomics Data Center (Nucleic Acids Res 2022), China National Center for Bioinformation / Beijing Institute of Genomics, Chinese Academy of Sciences (GSA: CRA015622), and are publicly accessible at https://ngdc.cncb.ac.cn/gsa. Please find lines 336-340 in the revised manuscript.

- in Line 130 and Line 145 is stated that KEGG enrichment analysis showed that the DEGs are enriched in auxin signal transduction pathway, tryptophan metabolism, oxidoreductase activity, hydrogen peroxide catabolic process and antioxidant activity. However, in Figure 2C and D (i.e., the top 20 in GO and KEGG analysis) only two of the above categories are shown. The authors should provide as a Suppl. Table all the enriched categories of their GO analysis

Thanks for the suggestion. We fully agree with the suggestion. As suggested, all the enriched categories of the GO analysis and KEGG analysis has been provided in Supplementary Table 3 and Supplementary Table 4. Please find lines 106-107 and lines 119-120.

- in Lines 150-152 the authors say that total ROS content was lower in the mutant compared to wild-type, in contrast to Figure 5A (the opposite is also stated in Lines 202-203). Additionally, I believe that the data in Lines 157-170 could be provided as a Table or Figure; giving all theses information in text may be tiring and confusing for the readers

Thank you for the critical comment. There was an error in lines 150-152. Therefore, we have corrected this sentence and please find lines 163-164 in the revised manuscript.

We fully agree with the suggestion. As suggested, the data has been provided in Figure 5, please find lines 184-188. The text has also been revised accordingly, please find lines 171-183.

- the authors should rewrite Lines 220-228 in order to be more comprehensive and to avoid misunderstanding. I also suggest that Suppl. Figure 2 could be a main figure and (as a model) to be more discussed. 

Thanks for your kind question. We fully agree with the suggestion. The discussion has been rewriten to be more comprehensive, please find lines 213-235.

We fully agree with the suggestion. As suggested, we revised Supplementary Figure 2 and put the model as a main figure, please find lines 247-250 (Figure 6).

Comments on the Quality of English Language

Minor editing of English language required

Thanks for your kind question. We fully agree with the suggestion. Therefore, we polished the manuscript with the help of an editing service and marked it in red in the revised manuscript.

Reviewer 3 Report

Comments and Suggestions for Authors

The presented article describes the influence of ZmNAC17 gene on maize mesocotyl elongation. With the use of transcriptomic sequencing 444 differentially expressed genes were identified. At the same time it was found, that content of indole acetic acid was significantly decreased in the zmnac17 mutant, indicating that the auxin metabolism pathway was changed in the zmnac17 mutant. Authors supposed that ZmNAC17 gene could regulate other auxin-related genes, inducing auxin synthesis and signaling pathway and affect mesocotyl elongation.

At the same time, the article needs significant improvement.

Description of the experiments must be presented in details (both plants growing and molecular analyses). Size of figures must be increased, because sometimes it is impossible to distinguish letters.

Line 25: What do you mean by “topsoil ability”?

Line 65:However, the function between NAC and mesocotyl elongation has not been well studied.”

Sentence is unclear.

Line 65 – 70. This is your result. It must be replaced.

Line 125 – 126. “In order to verify the relationship between auxin and mesocotyl elongation in zmnac 17-1 mutant.” – a fragment

Line 129 – 134. Sense is unclear.

Line 231-241. “Phenotypic analyses” is absolutely unclear.

Please describe in details: were seedlings grown in open air or not? Why plants were grown in different geographic locations, if you declare that experiment was carried out in the dark and so day light duration doesn’t matter?

Line 255-259. There is no information about un-etiolated seedlings.

Line 264 “...difference between the two groups...” Which groups?

Comments on the Quality of English Language

Line 82. ... “while in zmnac17-1 was about 0.23g.”

Line 165-166. “We also measured the content of GSH and GSSG. GSH content is 0.65 umol/g in B73 while is 0.75 umol/g in zmnac17-1, significantly elevated 15% in zmnac17-1 compared with B73.” – unclear.

Line 194....” to promotes lateral root development.”

Line 237 “...seeds were sowed in..”

Author Response

Thank you very much for taking the time to review this manuscript. Please find the detailed responses below and the corresponding revisions highlighted in Red in the re-submitted files.

The presented article describes the influence of ZmNAC17 gene on maize mesocotyl elongation. With the use of transcriptomic sequencing 444 differentially expressed genes were identified. At the same time it was found, that content of indole acetic acid was significantly decreased in the zmnac17 mutant, indicating that the auxin metabolism pathway was changed in the zmnac17 mutant. Authors supposed that ZmNAC17 gene could regulate other auxin-related genes, inducing auxin synthesis and signaling pathway and affect mesocotyl elongation.

At the same time, the article needs significant improvement.

Description of the experiments must be presented in details (both plants growing and molecular analyses). Size of figures must be increased, because sometimes it is impossible to distinguish letters.

Thank you very much for your recommendation. We fully agree with the suggestion. Therefore, we have polished the manuscript with the help of an editing service and marked it in red in the revised manuscript. We have modified the experimental method, please see lines 253-267, 277-292. We've updated some of the data.

Line 25: What do you mean by “topsoil ability”?

Thanks for your kind question. It has been revised to “germination ability”, please see lines 26 in the revised manuscript.

Line 65: “However, the function between NAC and mesocotyl elongation has not been well studied.”

Sentence is unclear.

Thanks for your kind question. It has been revised to “Despite the fact that extensive research has been conducted on the NAC family, its relationship with mesocotyl elongation has not been thoroughly studied.”, please see lines 68-69 in the revised manuscript.

Line 65 – 70. This is your result. It must be replaced.

Thanks for your kind question. We fully agree with the suggestion. It has been revised as suggested, please see lines 70-73 in the revised manuscript.

Line 125 – 126. “In order to verify the relationship between auxin and mesocotyl elongation in zmnac 17-1 mutant.” – a fragment

Thanks for your kind question. We fully agree with the suggestion. It has been revised to “In order to verify whether ZmNAC17 regulates the biosynthesis of auxin”, please see lines 133-134 in the revised manuscript.

Line 129 – 134. Sense is unclear.

Thanks for your kind question. We fully agree with the suggestion. It has been revised as suggested, please see lines 135-142.

Line 231-241. “Phenotypic analyses” is absolutely unclear.

Please describe in details: were seedlings grown in open air or not? Why plants were grown in different geographic locations, if you declare that experiment was carried out in the dark and so day light duration doesn’t matter?

Thanks for the critical comments. We fully agree with the suggestion. It has been revised as suggested. Plants were grown in different geographic locations for propagation and genotyping. Please see lines 253-267. 

Line 255-259. There is no information about un-etiolated seedlings.

Thanks for your kind question. For all of the experiments, we incubated seedlings in the dark. Therefore, there is no information about un-etiolated seedlings.

Line 264 “...difference between the two groups...” Which groups?

Thanks for your kind question. We fully agree with the suggestion. It has been revised to “DESeq2 was used to analyze the expression-related differences between B73 and zmnac17-1”, please see lines 301-302.

Line 82. ... “while in zmnac17-1 was about 0.23g.”

Thanks for your kind question. It has been revised to “while in zmnac17-1, it was about 0.23 g”, please see line 84.

Line 165-166. “We also measured the content of GSH and GSSG. GSH content is 0.65 umol/g in B73 while is 0.75 umol/g in zmnac17-1, significantly elevated 15% in zmnac17-1 compared with B73.” – unclear.

Thanks for your kind question. It has been revised to “The results show that GSH was 0.65 umol/g in B73 and 0.75 umol/g in zmnac17-1; i.e., the content of GSH was 15% higher in zmnac17-1.”, please see lines 171-172.

Line 194....” to promotes lateral root development.”

Thanks for your kind question. It has been revised to “AtNAC1 promotes lateral root growth by activating the expression of two auxin-related genes, DNA-binding protein (DBP) and auxin-induced in root cultures3 (AIR3)”, please see lines 207-209.

Line 237 “...seeds were sowed in..”

Thanks for your kind question. It has been revised to “Seeds were sown in”, please see line 263.

Round 2

Reviewer 3 Report

Comments and Suggestions for Authors

Authors have greatly improved the article. Now it can be published.

Author Response

Thank you very much for taking the time to review this manuscript and all the critical suggestions.